# Looking Beyond Single Images for Contrastive Semantic Segmentation Learning

**Feihu Zhang**     **Philip Torr**
University of Oxford

**René Ranftl**[*]     **Stephan R. Richter**[*]
Intel Labs

## Abstract

We present an approach to contrastive representation learning for semantic segmentation. Our approach leverages the representational power of existing feature extractors to find corresponding regions across images. These cross-image correspondences are used as *auxiliary labels* to guide the pixel-level selection of positive and negative samples for more effective contrastive learning in semantic segmentation. We show that auxiliary labels can be generated from a variety of feature extractors, ranging from image classification networks that have been trained using unsupervised contrastive learning to segmentation models that have been trained on a small amount of labeled data. We additionally introduce a novel metric for rapidly judging the quality of a given auxiliary-labeling strategy, and empirically analyze various factors that influence the performance of contrastive learning for semantic segmentation. We demonstrate the effectiveness of our method both in the low-data as well as the high-data regime on various datasets. Our experiments show that contrastive learning with our auxiliary-labeling approach consistently boosts semantic segmentation accuracy when compared to standard ImageNet pre-training and outperforms existing approaches of contrastive and semi-supervised semantic segmentation.

## 1   Introduction

Training semantic segmentation models typically requires a large amount of densely annotated images. Producing such dense annotations for an adequate number of images at sufficient quality is an extremely laborious and costly task [17]. Self-supervised and weakly-supervised learning techniques offer the possibility to significantly reduce the manual labeling effort that is required to train high-performance machine learning models. Contrastive learning [14, 22] has recently emerged as a promising technique for self-supervised representation learning that can leverage large amounts of unlabeled data for augmenting and extending existing labeled datasets without costly manual annotation. However, recent works on contrastive learning primarily focused on image/instance-level learning. Few works have studied contrastive learning in dense image labeling tasks, such as semantic segmentation [50, 51], as dense pixel-level segmentation poses unique challenges.

At its core, self-supervised contrastive learning is rooted in the availability of training samples that can be grouped automatically into *positive* pairs of similar concepts, and *negative* pairs of dissimilar concepts. The notion of similarity thereby depends strongly on the task at hand.

The appropriate selection of such pairs is of paramount importance for the performance of the resulting models. While it is often straightforward to generate negative pairs, automatically generating positive pairs can present a formidable challenge that requires intimate knowledge about the data and task at hand. Image augmentations are both simple and effective to generate positive pairs for image/instance-level learning [14, 15] but are too coarse-grained to learn representations at a pixel level.

---

[*]Equal advising.

35th Conference on Neural Information Processing Systems (NeurIPS 2021).

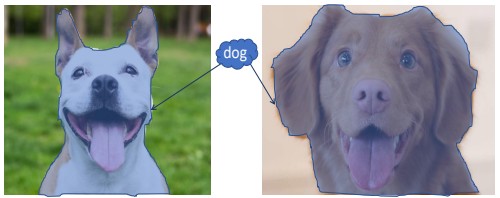 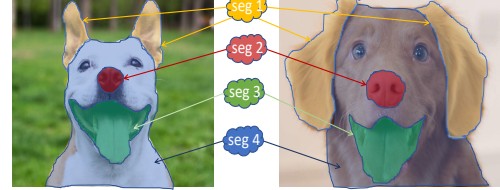

(a) Pseudo-labels for semi-supervised learning     (b) Auxiliary labels for our contrastive learning

Figure 1: Pseudo-labels vs. auxiliary labels. (a) Semi-supervised semantic segmentation requires pseudo-labels to be compatible with the ground truth labels. (b) Our method employs auxiliary labels to find corresponding regions across images. These correspondences do not need to match the ground truth classes exactly as we leverage contrastive learning.

Recent works [51, 56] adapt the idea of using image augmentations to generate positive pairs for more fine-grained pixel-level representation learning. While surprisingly effective, image augmentations have only a limited capability of covering the full visual spectrum of how two semantically related pixels or patches can look. Mid and high-level information such as context, texture, large visual shifts inside a semantic class (for example, the class "car" may comprise a wide range of different shapes and colors), and even projective distortions due to the imaging process, may be extremely relevant but cannot be simulated realistically with augmentations alone. As a consequence, methods that exclusively rely on augmentations are limited in the contrastive information they can provide to downstream tasks. A different line of work thus exploits ground truth semantic labels to establish correspondences between images to generate more diverse and informative positive pairings [50, 52]. However, the need for labeled data impedes scaling these approaches to the massive amounts of data required to fully reap the benefits of contrastive learning [14, 22].

In this work, we explore the generation of *auxiliary labels* to establish fine-grained, pixel-level correspondences across images to select contrastive pairs. We distinguish auxiliary labels from the more commonly employed pseudo-labels [66], as auxiliary labels need not represent the same classes as the ground truth labels. This concept is illustrated in Figure 1. To this end, we leverage feature embeddings that we generate with existing feature extractors. The feature extractors have either been trained using coarse image-level supervision, on a small set of pixel-level labeled data, or using image-level, self-supervised contrastive learning approaches [16, 22]. We cluster feature embeddings and assign auxiliary labels to individual pixels based on cluster membership. We also introduce a simple, yet effective, voting strategy to spatially regularize the auxiliary label maps. The auxiliary labels tend to cluster into semantically and perceptually meaningful groups which can be used to establish cross-image correspondences. This allows us to generate a diverse and realistic set of positive pairs that can be used for fine-grained contrastive visual representation learning. An example that illustrates our approach is shown in Figure 2.

We additionally introduce a proxy metric that allows us to quickly assess the quality of any auxiliary labeling without the need to train a model. Finally, we present a comprehensive experimental

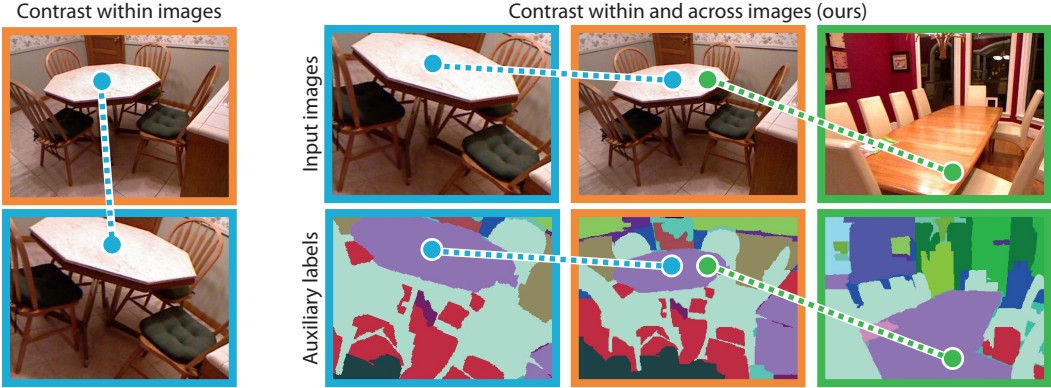

Figure 2: Comparison of augmentation-based contrastive learning (left) to our proposed cross-image contrastive learning with auxiliary labels (right). For an anchor image, prior work [51, 56] samples positive correspondences only within augmentations of the same image. Our method additionally establishes correspondences to other images based on matching auxiliary labels.

evaluation of various factors and hyper-parameters that influence the performance of contrastive training in semantic segmentation.

Our experiments across three datasets and two network architectures show that contrastive learning with auxiliary labels consistently boosts semantic segmentation performance when compared to standard ImageNet pre-training and outperforms existing approaches for contrastive and semi-supervised semantic segmentation.

## 2 Related Work

Approaches to self-supervised learning via a contrastive learning objective have recently demonstrated impressive performance compared to fully supervised methods in a wide variety of tasks [14–16, 22, 24]. In self-supervised learning, the contrastive objective is used as a pretext task to learn general visual representations, which can be well transferred to downstream tasks via fine-tuning. The intuition of the contrastive objective is straightforward: similar visual concepts should be mapped to nearby representations in latent space, whereas dissimilar concepts should be mapped further apart. A crucial ingredient is the definition of similarity and hence the selection of similar and dissimilar samples during training. Seminal work adopted instance discrimination [14, 22]. That is, positive samples are generated from the same image via different transformations (views). All other images available for training represent potential negative samples. Recent work investigated different strategies to generate more informative positive and negative pairs, including mutual view selection [46, 47], hard mining [30, 42, 50], adversarial perturbations [25, 26, 29, 32], and using large mini-batches or a momentum memory bank [22, 37, 53].

Another line of work exploited ground truth labels to find positive and negative pairs for contrastive learning [6, 31, 49, 50, 52, 61]. While this approach may improve performance in fully supervised settings, it cannot be applied to unlabeled supplementary data.

**Contrastive learning for segmentation.** Recent works adapted contrastive learning to dense prediction tasks such as semantic segmentation [10, 51, 56, 65]. DenseCL applied the contrastive objective to the pixel level [51]. ReSim learned spatial features from sliding windows [54]. Xie et al. generated views by smoothing over nearby feature representations to aggregate local context [56]. Chen et al. exploited foreground/backward masks [11] while Van Gansbeke et al. leveraged object masks from pre-trained saliency detectors to group samples [48]. All of the above-mentioned approaches generate positive views from the same image. This limits the diversity of positive samples severely, as instances of the same category in different images cannot be paired. In contrast to existing works, our auxiliary-labeling strategy allows us to establish positive correspondences across images and consequently to pair more diverse examples of the same concept.

**Pseudo-labels in semi-supervised semantic segmentation.** Pseudo-labels are a surrogate ground truth for unlabeled data. Many existing works [2, 12, 35, 63, 63, 65] train a teacher network on labeled data in order to produce surrogate annotations on the unlabeled data available. A student network is then trained on both the ground truth and the pseudo-labels. The surrogate annotations are usually far from perfect, forcing the student to learn from noisy (and sometimes just misleading) data, which may negatively affect its performance [3, 45]. An underlying problem in this setup is the need for compatibility between pseudo labels and ground truth labels. This forces a (possibly wrong) decision on the teacher network, while potentially more fine-grained knowledge present in the feature activations used to make the decision is completely ignored [39].

The auxiliary labels we construct from feature extractors exploit exactly this type of knowledge. By clustering feature representations, we obtain a more fine-grained set of labels that does not need to match the ground truth label set. Besides providing a more comprehensive perspective on the unlabeled data, auxiliary labels also allow us to leverage a wider range of feature extractors than just a (teacher) method built for the same task.

**Unsupervised clustering for representations learning.** Several works proposed unsupervised clustering objectives for deep representation learning, including approaches to image or instance-level representation learning [5, 21, 27, 28, 55, 57, 58, 58, 64] as well as image segmentation [28, 33, 38]. A line work clusters features and uses the cluster assignments as pseudo-labels for self-training [4, 7–9]. While we generate auxiliary labels also by clustering features, we leverage powerful existing feature

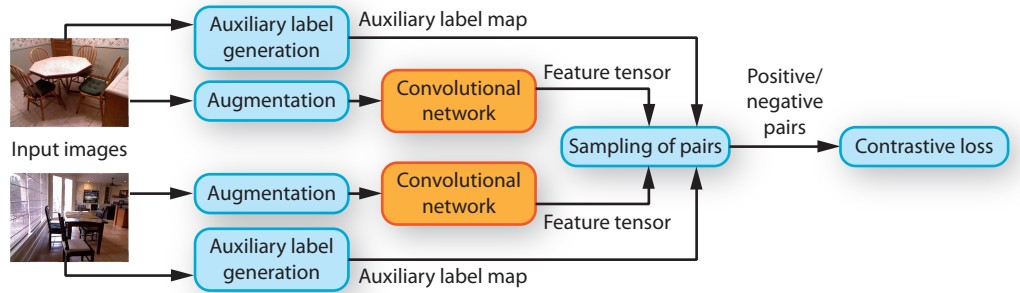

Figure 3: Method overview. Our method augments input images and feeds them to a convolutional network. We generate an auxiliary label map for each image and use it to guide the sampling of contrastive pairs. The network is trained using a confidence-weighted contrastive loss. Trainable components are shown in orange.

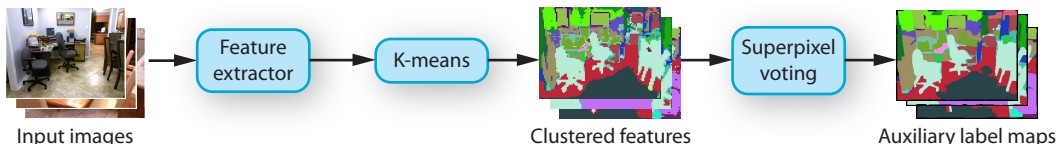

Figure 4: We generate auxiliary labels using existing feature extractors. We cluster features from the whole dataset with K-means. Pixels are assigned to clusters based on their distance to the cluster centers. A final smoothing step using superpixel voting aligns the auxiliary labels to image edges and removes outliers.

extractors instead of a self-trained network. Furthermore, instead of learning image-level visual representations, we address the challenges of dense prediction tasks, such as semantic segmentation.

## 3   Method

The key to contrastive learning is the definition of positive and negative pairs, i.e., which samples should share a representation, and which samples should be mapped to different representations. For annotated datasets, the decision on how to construct positive and negative pairs is clear: samples with matching labels define the positives while samples with different labels define the negatives [31,50,52]. However, the need for annotations severely limits the data available for learning.

Missing any information about correspondences in the dataset, unsupervised approaches take a conservative stance and implicitly assume all randomly paired samples to mismatch. Matching samples are synthesized by sampling a single image and transforming it in different ways. The resulting views represent positive correspondences for training [14–16, 22, 24]. An obvious problem with this approach is that correct pairs (e.g., the same car in different images) will be labeled as negative. At the same time, invariances are only learned w.r.t. the simple transformations applied.

To mitigate this issue, we generate auxiliary labels that allow establishing positive pairs across the whole training set akin to supervised methods. However, our auxiliary labels are generated using existing feature extractors, which alleviates the need for dense annotations and unlocks training on large-scale unlabeled data. Figure 3 provides an overview of our method.

**Auxiliary label generation.**   Our approach for generating auxiliary labels consists of three steps: From a set of unlabeled images, we obtain visual features with an existing feature extractor. We cluster the features across the whole dataset and assign individual pixels in the input images to their corresponding cluster center. In a final step, we refine the resulting label maps through a super-pixel voting algorithm. A schematic overview of our approach is shown in Figure 4.

In principle, any feature extraction method can be used for generating auxiliary labels and we study several choices. First, we employ a ResNet-50 [23], trained through self-supervised contrastive learning [22]. It represents a feature extractor that has seen no manual annotations. Second, we use the same ResNet-50 architecture but trained for image classification on the ImageNet dataset

to study the influence of coarse, image-level labels for training the feature extractor. Finally, we explore several instances of a DeepLabV3 [13] semantic segmentation network that has been trained for semantic segmentation on various datasets.

When using ResNet, we extract features before the final adaptive average pooling layer. This yields a representation at 1/32 of the input resolution. We extract multi-scale features by rescaling the input image to various resolutions and passing it to the feature extractor. We concatenate the feature maps from multiple scales, upsample the representation to 1/8 of the input image size through bilinear interpolation, and normalize the resulting feature vectors using $L_2$ normalization. For DeepLabV3 feature extractors, we extract features before the penultimate layer (counting the final linear layer as the last layer) and follow the same multi-scale feature extraction procedure as before.

We apply our feature extraction approach to all images in an unlabeled dataset and generate a set of auxiliary classes using K-means clustering [40]. Since the total number of features in an unlabeled dataset can be extremely large (on the order of tens or hundreds of millions of feature vectors), we perform the clustering step on features from $5\,000$ images that we sample uniformly at random. We annotate the remaining images in the dataset by assigning pixels to the closest auxiliary class in terms of the $L_2$ distance between the pixel feature vector and the cluster center. To remove outliers and false auxiliary labels, we drop the 5% pixels with the largest distance to the nearest cluster center. An example result of our labeling strategy is shown in Figure 2.

The resolution and spatial consistency of auxiliary labels depends on the feature extractor employed. Some may produce only noisy or low-resolution feature maps. We thus refine the auxiliary label maps with a simple superpixel voting strategy. We compute a superpixel over-segmentation [1] and assign the most common label within a superpixel to all pixels of the superpixel. This aligns label map boundaries to image edges, removes outliers, and alleviates errors from low-resolution features.

**Contrastive training.** We use the auxiliary labels to generate positive and negative pairs both within an image as well as across images, and leverage them for pre-training semantic segmentation models. We train two standard segmentation architectures in our experiments [13, 60]. The final classification layer of both models is replaced with a linear layer that outputs a 128-dimensional feature map at 1/8 of the input resolution. For each training iteration, we randomly sample $10\,000$ feature vectors as anchor vectors $\boldsymbol{v} \in \mathbb{R}^{128}$ and a different set of $40\,000$ feature vectors as contrastive feature candidates ($\boldsymbol{v}^+ \in \mathbb{R}^{128}$ or $\boldsymbol{v}^- \in \mathbb{R}^{128}$). All anchors are selected from the current mini-batch, whereas we select $10\,000$ contrastive features from the current mini-batch and the remaining $30\,000$ contrastive features from a memory bank [22]. For every anchor $\boldsymbol{v}$, we construct a set of associated positive pairs $\mathcal{P}(\boldsymbol{v}) = \{(\boldsymbol{v}, \boldsymbol{v}^+)\}$ from all contrastive candidates where the auxiliary label of a candidate matches the anchor. We similarly create the set of negative pairs $\mathcal{N}(\boldsymbol{v}) = \{(\boldsymbol{v}, \boldsymbol{v}^-)\}$ from all contrast feature vectors where the auxiliary labels differ.

Our training loss is inspired by the supervised contrastive loss of Khosla et al. [31]. We define the contrastive distance for a single pair as

$$\mathcal{L}(\boldsymbol{v}, \boldsymbol{v}^+) \quad = \quad -\log \frac{\exp(\boldsymbol{v} \cdot \boldsymbol{v}^+/\tau)}{\exp(\boldsymbol{v} \cdot \boldsymbol{v}^+/\tau) + \sum_{(\boldsymbol{v}, \boldsymbol{v}^-) \in \mathcal{N}(\boldsymbol{v})} \exp(\boldsymbol{v} \cdot \boldsymbol{v}^-/\tau)}, \quad (1)$$

where we set the temperature $\tau = 0.07$ in our experiments. To compute the loss over all pairs, we introduce a confidence weight that is based on the expected reliability of a positive pair. To this end, we partition the set of positive pairs into three subsets: the set $\mathcal{P}_0$ consists of pairs for which both the anchor and the contrast sample belong to the same superpixel. We assign particularly high confidence to these pairs due to the inherent high spatial regularity of superpixels and feature extractors. We assign medium weight to the set $\mathcal{P}_1$ that consists of pairs sampled from the same image. Finally, the set $\mathcal{P}_2$ consists of samples where the anchor and contrastive features are from different images. Compared with samples in $\mathcal{P}_0$ and $\mathcal{P}_1$, these cross-image samples are more likely to be wrongly labeled as positive pairs. We thus assign the lowest weight to these samples. Our final loss is given by

$$\mathcal{L} \quad = \quad \frac{\sum_{i=0}^{2} \lambda_i \sum_{(\boldsymbol{v}, \boldsymbol{v}^+) \in \mathcal{P}_i} \mathcal{L}(\boldsymbol{v}, \boldsymbol{v}^+)}{\sum_{i=0}^{2} \lambda_i |\mathcal{P}_i|}, \quad (2)$$

where $\lambda_{0,1,2} = \{10, 4, 1\}$ for all our experiments.

**Auxiliary label quality.** Our label generation strategy supports various technical choices that affect the efficacy of the auxiliary label maps. For example, different feature extractors or a different

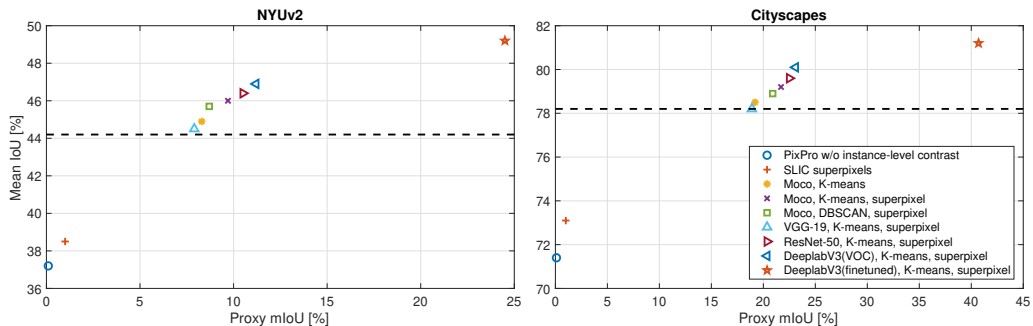

Figure 5: Auxiliary label quality and final segmentation performance for conditions in our controlled experiments. The proposed proxy mIoU correlates strongly with the mIoU of segmentations after fine-tuning. Dashed lines mark the performance of a DeeplabV3 baseline with ImageNet pre-training.

number of auxiliary classes in the clustering step will produce auxiliary labelings that will lead to different performance for a given downstream task. We introduce a novel metric that can be quickly computed and can serve as a guiding metric to choose a specific auxiliary-labeling strategy when a representative set of labeled images is available.

Let $S = \{S_j\}_{j=1}^K$ be the set of auxiliary segments, where $S_j$ is defined as the set of pixels with the auxiliary label $j$ (computed over the complete validation set). Similarly, let $G = \{G_i\}_{i=1}^M$ be the set of true segments, where $G_i$ is the set of pixels that correspond to the ground truth class $i$. We define the *Proxy mean Intersection-over-Union (PmIoU)* as

$$\text{PmIoU}(S, G) = \frac{1}{\max\{K, M\}} \sum_{S_j} \max_{G_i} \left\{ \frac{|S_j \cap G_i|}{|S_j \cup G_i|} \right\}. \tag{3}$$

The metric assigns each auxiliary segment $S_j$ to the ground truth segment $G_i$ that yields the highest Intersection-over-Union (IoU) among all possible assignments, as we do not know the correct correspondence between auxiliary labels and ground truth labels (additionally, in most cases a one-to-one correspondence will not exist). Intuitively, PmIoU reflects the highest mIoU that is achievable by optimally matching the auxiliary labels with the ground truth classes. In the ideal case, where the auxiliary labels exactly coincide with the ground truth labels, the PmIoU equals one. On the other hand, if the auxiliary labels represent a strong over-segmentation or are crossing spatial boundaries between different ground truth segments, the PmIoU will be small.

Figure 5 shows PmIoU together with the mIoU after fine-tuning a model that was pre-trained with the corresponding auxiliary labels. We observe that PmIoU correlates well with mIoU across auxiliary labeling strategies. This indicates that PmIoU is an excellent and, most importantly, computationally cheap metric to evaluate different auxiliary-labeling strategies.

**Practical considerations.** In practice, we find that a large amount of positive pairs improves convergence speed, while a sufficiently large amount of negative pairs are required to train models that perform well on the downstream task. We thus implement a memory-efficient version of our loss that doesn't explicitly store the inner products between all pairs. This allows us to scale the total number of pairs per mini-batch substantially. With $10\,000$ anchor feature vectors and $40\,000$ contrast feature vectors, we can, depending on the composition of the mini-batch and auxiliary labels, see on average 10 million positive pairs in a single mini-batch. We find that with this very large sample size, 20 epochs suffice to train strong representations for segmentation models. For comparison, MoCo [22] uses only 256 positive pairs per mini-batch and consequently requires more than 200 training epochs.

Supervised semantic segmentation models commonly employ an auxiliary loss on mid-level features to improve performance [13, 60]. We find that an auxiliary contrastive loss on mid-level features can likewise improve the representational power of our contrastively trained models. State-of-the-art semantic segmentation models output features at a lower resolution than the input image (usually 1/8 of the input size). We train with the contrastive loss both at the low output resolution that is defined by the model, as well as at the original resolution of the input image (by upsampling the feature maps). We give equal weight to both losses during training.

In addition to large mini-batches, momentum contrast [22] can provide candidate pairs with high diversity and is commonly employed for effective contrastive learning. We also adopt this technique.

## 4 Evaluation

We evaluate our approach on multiple challenging datasets: NYUv2 [44] contains about $500\,000$ unlabeled and $1449$ labeled images of indoor scenes. The labeled set is annotated with $40$ classes and split into 795 train and 654 test images. Cityscapes [17] contains more than $100\,000$ unlabeled video frames of driving scenes, 2975 labeled images for training and another 500 annotated images for validation. ADE20K [62] consists of $20\,000$ training and 2000 validation images, all of which are fully annotated. Due to the lack of unlabeled images, we supplement ADE20K with MS-COCO [36] (without using MS-COCO's annotations).

For generating auxiliary labels from multi-scale feature maps, we resize input images to the ResNet feature extractor by factors 1, 2, 4 for NYUv2 & ADE20K, and by 0.5, 1, 2 for Cityscapes. With a DeepLabV3 feature extractor, we scale input images for all datasets by factors 0.5, 0.75, 1, 1.5, 2. We generate $K = 1.5M$ auxiliary labels, where $M$ is the number of groundtruth classes in the target dataset. We used our proposed PmIoU proxy metric to determine this number.

Our approach can be used to train any semantic segmentation architectures. We choose two popular architectures, DeepLabV3 [13] and PSPNet [60][2], both with a ResNet-50 backbone for our experiments. All experiments were conducted on 8 Nvidia Quadro 6000 GPUs. We use batches of 128 input images and set the crop size to $321 \times 361$. The memory bank for implementing momentum contrast holds the features of $384$ frames. All models are trained using SGD with momentum [43]. We set weight decay to $1e-4$ and the momentum term to $0.9$. We perform warmup with a learning rate of $0.01$ for two epochs and then train for another 18 epochs with a learning rate of $0.1$. The learning rate is reduced by a factor of 10 after epochs 10, 15, and 18, respectively. We use the augmentations defined in SimCLR [14], which include random scaling, rotation, cropping, and color transformations, together with CutOut [18], where we fill the cut-out region with the mean value of the cut-out. A single training run on NYUv2 takes approximately 50 hours.

For fine-tuning, we use the same hyper-parameters as defined in the original works [13, 60].

|  | Method | NYUv2 | | | Cityscapes | | | ADE20K | | |
|---|---|---|---|---|---|---|---|---|---|---|
|  |  | 10% | 20% | 100% | 10% | 20% | 100% | 10% | 20% | 100% |
|  | Random initialization | - | - | 25.3 | - | - | 68.2 | - | - | 35.1 |
|  | ImageNet | 20.4 | 30.3 | 44.2 | 42.3 | 57.2 | 78.2 | 20.2 | 26.7 | 43.0 |
| Baselines | SWAV [9] | - | - | 41.2 | - | - | 77.3 | - | - | 43.1 |
|  | PCL [34] | - | - | 43.9 | - | - | 77.6 | - | - | 42.9 |
|  | MoCo-v2 [16] | 19.1 | 29.5 | 43.8 | 41.5 | 56.3 | 77.9 | 19.7 | 26.3 | 42.8 |
|  | DenseCL (NYUv2) [51] | 17.2 | 26.1 | 39.8 | - | - | - | - | - | - |
|  | DenseCL (ImageNet) [51] | - | - | 44.8 | - | - | 78.6 | - | - | 43.4 |
|  | PixPro [56] | 22.1 | 31.2 | 45.0 | 43.5 | 58.1 | 78.4 | 21.4 | 27.2 | 43.2 |
|  | PseudoSeg [66] | 30.8 | 35.9 | 47.1 | 48.1 | 59.7 | 79.9 | - | - | 43.6 |
| Ours | ResNet-50 (MoCo) | 29.8 | 35.1 | 46.0 | 48.2 | 60.1 | 79.2 | 24.7 | 29.1 | 43.7 |
|  | ResNet-50 (ImageNet) | 30.4 | 35.7 | 46.4 | 48.4 | 60.3 | 79.6 | 25.2 | 29.7 | 43.9 |
|  | DeepLabV3 (Pascal VOC) | 31.7 | 36.3 | 46.9 | 49.1 | 60.0 | 80.1 | 25.7 | 30.0 | 44.0 |
|  | DeepLabV3 (VIPER) | - | - | - | 48.1 | 60.2 | 79.8 | - | - | - |
|  | DeepLabV3 (*) | **36.1** | **39.4** | **49.2** | **52.1** | **61.9** | **81.2** | **27.1** | **31.2** | **44.5** |

Table 1: Comparison to prior work. We report mIoU after fine-tuning on various fractions (10%, 20%, 100%) of the target dataset. We compare to several contrastive learning methods [16, 51, 56] and to a state-of-the-art semi-supervised semantic segmentation model [66]. To generate auxiliary labels we employ feature extractors trained for image classification using self-supervised learning (MoCo) and labeled data (ImageNet), or semantic segmentation networks that were trained on another dataset (Pascal VOC [20]), on synthetic data (VIPER [41]), or on the target dataset (*).

---

[2]Results for PSPNet are available in supplementary materials.

| Method | Pixel-level contrast | Cross-image contrast | Epochs | mIoU |
|---|:---:|:---:|---:|---:|
| MoCo-v1 [22] | ✗ | ✗ | 80 | 42.1 |
| MoCo-v1 [22] | ✗ | ✗ | 200 | 47.0 |
| Hierarchical Grouping [59] | ✓ | ✗ | 80 | 46.5 |
| Ours (MoCo) | ✓ | ✓ | 20 | 49.1 |
| Ours (*) | ✓ | ✓ | 20 | **54.3** |

Table 2: Comparison to prior work [59] on the Pascal VOC validation set. All methods were pre-trained on a mix of the Pascal VOC [20] and MS-COCO [36] training sets (without ImageNet pre-training). Our approach outperforms the baselines both with an unsupervised feature extractor (MoCo) and with a semi-supervised feature extractor (*) with only a quarter of training epochs.

**Comparison to prior work.** We first compare to prior work on unsupervised and semi-supervised representation learning. We compare to different state-of-the-art contrastive learning methods, including image-level contrastive learning [22] and pixel-to-pixel dense contrastive learning methods [51, 56]. Our results with different feature extractors are shown in Table 1. Existing contrastive learning methods (sometimes implicitly) assume that only a single object is present in a training image. We thus find that they struggle to learn competitive representations when trained on the complex images that are typically found in semantic segmentation datasets. For example, when training DenseContrast [51] on the NYUv2 dataset, we observe that the results trail ImageNet pre-training by more than 4%. We see much stronger performance of this approach when training on ImageNet data, where the results even slightly surpass training with labeled data. We consequently train the remaining contrastive baselines, MoCo and PixPro, only on ImageNet as they are subject to the same considerations.

Our method is specifically designed for semantic segmentation. It can be directly trained on complex semantic scenes and can consequently reap the benefits of seeing this more complex data. It consistently outperforms ImageNet pre-training and existing contrastive approaches across datasets, independent of the amount of total labeled samples that are given. We additionally compare our approach to PseudoSeg [66], a state-of-the-art semi-supervised segmentation model that leverages pseudo-labels and has been jointly trained on the same labeled and unlabeled data as our approach. Using all available labeled data, our method outperforms PseudoSeg by 2.1% mIoU on NYUv2, by 1.3% mIoU on Cityscapes, and by 0.9% mIoU on ADE20K, respectively. When only a fraction of labeled data is available, our approach provides even further gains over the state of the art. Figure 6 shows example auxiliary labels together with our results after fine-tuning on these datasets.

We additionally compare our approach to a closely related work that leverages hierarchical grouping to sample pixel-level positive pairs within a single image [59] in Table 2. We follow the protocol

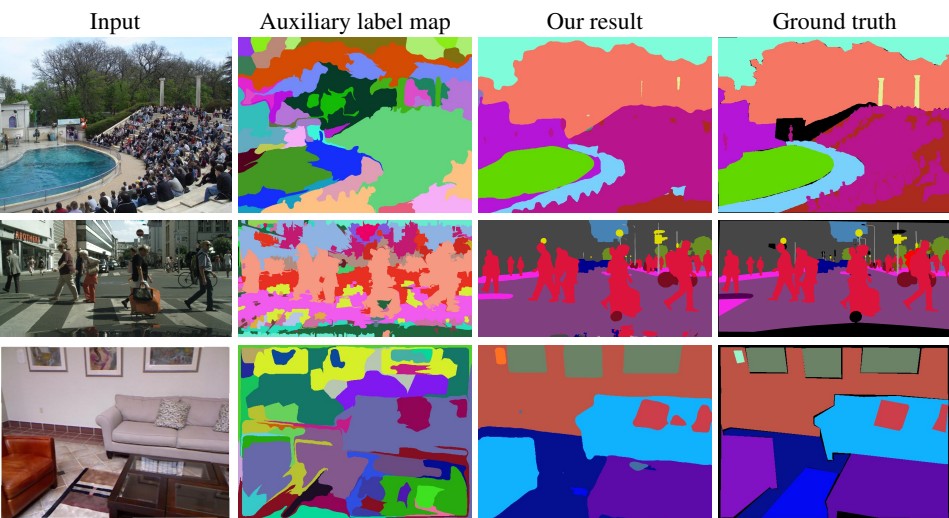

|  Input  |  Auxiliary label map  |  Our result  |  Ground truth  |

Figure 6: Comparison of auxiliary labels (unsupervised ResNet-50 (MoCo) as the feature extractor) and results after fine-tuning for samples from ADE20K, Cityscapes, and NYUv2 (top to bottom).

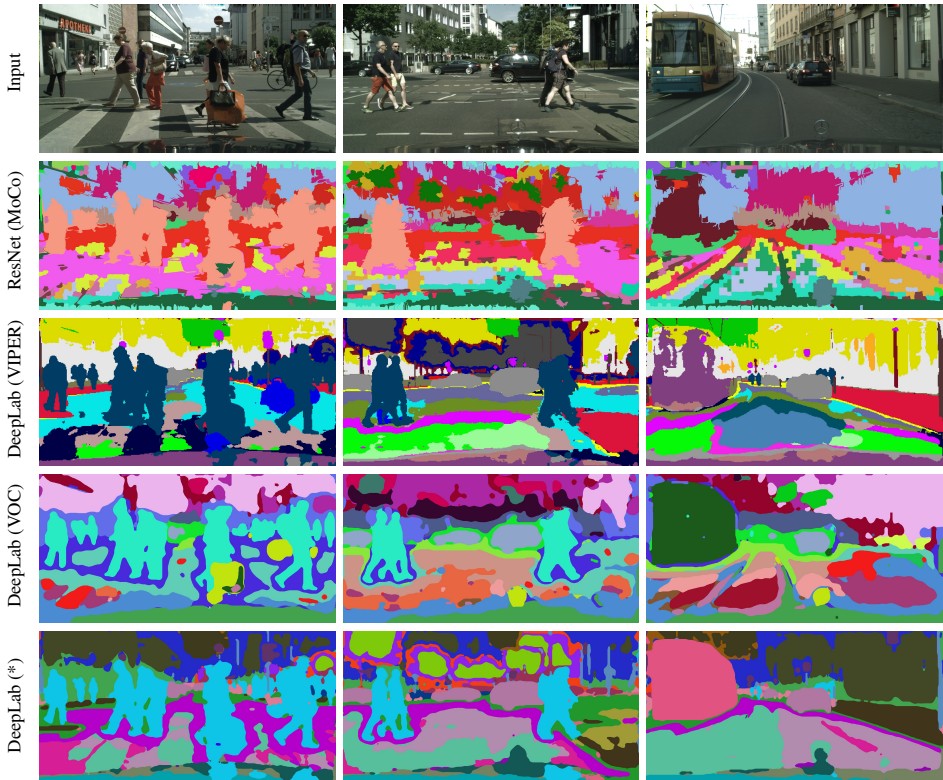

Figure 7: Qualitative comparison of auxiliary labels from different feature extractors. Corresponding colors indicate corresponding auxiliary labels across images.

of [59] and use a ResNet-50 backbone. All approaches (and feature extractors used in our method) were pre-trained on a mix of the Pascal VOC [20] and MS-COCO [36] training sets before fine-tuning on the Pascal VOC training set. Cross-image positive samples strongly improve performance even after only training for a quarter of epochs. The performance of [59] is comparable to MoCo-v1 [22], whereas our method outperforms the more advanced MoCo-v2 [16] baseline on other datasets.

**Feature extractors.** We assess how different feature extractors affect the segmentation performance after fine-tuning. We evaluate two ResNet-50 feature extractors, one of which was trained using unsupervised image-level contrastive learning on ImageNet data (without annotations), and one of which was trained on ImageNet labels. We further evaluate a DeepLabV3 feature extractor that has either been trained on Pascal VOC [20], the synthetic VIPER dataset [41], or on the target dataset itself. These feature extractors represent a spectrum of different labeling efforts that are required to construct them, ranging from no effort (MoCo) to potentially high effort when labeled data on the target dataset (*) is required. Table 1 (bottom) summarizes our results. While all variants yield strong results, we see that feature extractors that were provided with more informative labels and data that corresponds closely to the target task yield the best performance. Surprisingly, even the MoCo feature extractor, which was trained without any labeled data, yields results competitive with feature extractors that have been trained on ImageNet or even segmentation datasets (Pascal VOC and VIPER). We observe the best performance when training the feature extractor on the target dataset. Note that the results are consistent even when the amount of labeled data is reduced. Figure 7 shows example auxiliary labelings with different feature extractors.

**Controlled experiments.** We study the influence of different technical choices for generating auxiliary labels and report results in Figure 5 (corresponding numerical results are provided in supplementary materials). We find that superpixel refinement improves the auxiliary label quality and consequently leads to an improvement of the mIoU by 0.7-1.1%. We evaluate DBSCAN [19] as an alternative clustering algorithm but find that it performs slightly worse than K-means clustering while having higher memory and computational requirements.

| +&- selection | | Hyperparameters | | | | | | Settings of contrastive loss | | | mIoU |
|---|---|---|---|---|---|---|---|---|---|---|---|
| Anchors | Contrast | CutOut | Batch size | Crop size | MoCo | Data | Epochs | Multi-scale | Aux. loss | Confidence | |
| 100 | 400 | - | 32 | 321×361 | - | 20k | 10 | - | - | - | 39.2 |
| 1k | 4k | - | 32 | 321×361 | - | 20k | 10 | - | - | - | 42.1 |
| 10k | 40k | - | 32 | 321×361 | - | 20k | 10 | - | - | - | 42.7 |
| 10k | 40k | | 128 | 321×361 | - | 20k | 10 | - | - | - | 43.2 |
| 10k | 40k | √ | 128 | 321×361 | - | 20k | 10 | - | - | - | 43.5 |
| 10k | 40k | √ | 64 | 473×473 | - | 20k | 10 | - | - | - | 43.3 |
| 10k | 40k | √ | 128 | 321×361 | 384 | 20k | 10 | - | - | - | 43.7 |
| 10k | 40k | √ | 128 | 321×361 | 384 | 20k | 20 | - | - | - | 43.9 |
| 10k | 40k | √ | 128 | 321×361 | 384 | 128k | 20 | - | - | - | 45.0 |
| 10k | 40k | √ | 128 | 321×361 | 384 | 128k | 20 | √ | - | - | 45.2 |
| 10k | 40k | √ | 128 | 321×361 | 384 | 128k | 20 | √ | √ | - | 45.6 |
| 10k | 40k | √ | 128 | 321×361 | 384 | 128k | 20 | √ | √ | √ | 46.0 |
| 10k | 40k | √ | 128 | 321×361 | 384 | 480k | 40 | √ | √ | √ | 47.1 |

Table 3: Controlled experiments. "Anchors" and "Contrast" are the numbers of pixels that are selected as anchor features and contrast candidate features per iteration. "Multi-scale" denotes evaluating the contrastive loss on both 1/8 resolution and full resolution of the input image. "Aux. loss" adds a contrastive loss to an auxiliary middle layer, similar to the auxiliary loss used for fine-tuning DeeplabV3. "MoCo" denotes wether we use momentum contrast training [22].

We finally provide a comprehensive evaluation of the influence of various hyper-parameters. We show results on the NYUv2 dataset for these experiments and use a ResNet-50 trained with MoCo [16] on the unlabeled portion of the dataset to generate auxiliary labels. Evaluations are conducted on the validation set. We study the effects of (1) the number of positive pairs and negative pairs in a training iteration, (2) batch size and the momentum-based memory bank [22], (3) multi-scale contrastive training, (4) the influence of the auxiliary contrastive loss, and (5) the influence of the unlabeled training set size and the number of epochs. Table 3 provides an overview of our results. We find that selecting more positive and negative pairs (top section) in a training step improves mIoU and reduces the required number of epochs to get well-performing representations. CutOut augmentation has a positive influence and using more unlabeled data and training for more epochs further improves results (middle section). Increasing the crop size reduces performance, as it necessitates a reduction of the batch size and thus reduces the diversity of input samples due to memory constraints. The last section shows that multi-scale training slightly improves accuracy by 0.2% mIoU, while using an auxiliary contrastive loss on the mid-level features contributes another 0.4% improvement. Finally, using the proposed confidence weighting of the loss function improves accuracy by another 0.4%.

## 5 Discussion

In this paper, we study the use of auxiliary labels as guidance to build cross-image correspondences for contrastive learning in semantic segmentation. Our method leverages existing feature extractors to generate auxiliary labels. The requirements on the feature extractors are loose, as we demonstrate by using a broad set, ranging from networks trained for image classification to semantic segmentation approaches trained on synthetic data. For all feature extractors, our method consistently outperforms pre-training on ImageNet. Our exploration was guided by a new quality metric for auxiliary labels (Proxy mIoU), which can provide an estimate for the final segmentation performance after fine-tuning without the need for training a network. Finally, we propose several strategies for effective contrastive learning in semantic segmentation. We believe that our approach opens up new ways of integrating unlabeled training data for semantic segmentation models.

**Limitations.** Our method is demanding in its hardware requirements and its performance scales with available GPU memory. It shares this limitation with other state-of-the-art methods for contrastive learning [14, 22]. However, the computational cost of exploring new auxiliary-labeling strategies can be reduced substantially by applying our proposed proxy quality metric.

**Potential risks.** Every method that learns from data carries the risk of introducing biases. Additionally, the evaluation on specific benchmarks may suffer from dataset or concept bias. Work that bases itself on our method should carefully consider the consequences of any potential underlying biases.

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
