# Looking Beyond Single Images for Contrastive Semantic Segmentation Learning – Supplementary Material –

## 1   Additional results

### 1.1   Controlled experiment on auxiliary label generation

Table 1 reports the results of a controlled experiment evaluating different components in our framework for auxiliary label generation. We study different feature extractors, the use of an alternative clustering algorithm[1], and the influence of super-pixel voting.

| | Auxiliary label generation | | | NYUv2 | | Cityscapes | |
|---|---|---|---|---|---|---|---|
| # | Feature extractors | Clustering | Superpixel | PmIoU | mIoU | PmIoU | mIoU |
| 1 | - | - | - | <0.1 | 37.2 | <0.1 | 71.4 |
| 2 | - | - | $\checkmark$ | <1 | 38.5 | <1 | 73.1 |
| 3 | ResNet-50 (MoCo) | K-means | - | 8.3 | 44.9 | 19.2 | 78.5 |
| 4 | ResNet-50 (MoCo) | K-means | $\checkmark$ | 9.7 | 46.0 | 21.7 | 79.2 |
| 5 | ResNet-50 (MoCo) | DBSCAN | $\checkmark$ | 8.7 | 45.7 | 20.9 | 78.9 |
| 6 | VGG-19 (ImageNet) | K-means | $\checkmark$ | 7.9 | 44.5 | 18.9 | 78.2 |
| 7 | ResNet-50 (ImageNet) | K-means | $\checkmark$ | 10.5 | 46.4 | 22.5 | 79.6 |
| 8 | DeepLabV3 (Pascal VOC) | K-means | $\checkmark$ | 11.2 | 46.9 | 23.1 | 80.1 |
| 9 | DeepLabV3 (*) | K-means | $\checkmark$ | 24.5 | 49.2 | 40.7 | 81.2 |

Table 1: Ablation study of our auxiliary label generation framework. We report the proxy mean IoU (PmIoU) of the auxiliary labels on the validation set together with the mIoU after pre-training and fine-tuning. This data corresponds to Figure 5 in the main paper.

The first condition (#1) establishes a baseline using only intra-image contrast without any auxiliary labels. Positive correspondences are generated by matching pixels across different augmentations of the same image. In the second condition (#2) we over-segment each image. Here, positive correspondences are established between pixels originating from the same superpixel. The remaining conditions vary the feature extractors, the clustering algorithm, and the use of superpixels. We find that using over-segmentations for establishing correspondences consistently improves performance, be it without using any feature extractors (#1 vs. #2), or with auxiliary labels (#3 vs. #4). With respect to the clustering algorithm, K-means performs better than DBSCAN (#4 vs. #5), which is why we employ K-means in all remaining experiments. For the feature extractors used for generating auxiliary labels, we find that classifiers pre-trained on ImageNet beat pre-training with Moco (#7 vs. #4), and stronger backbones expectedly improve the auxiliary labels (#7 vs. #6). Furthermore, we find that feature extractors trained for the task at hand (here semantic segmentation) improve performance (#8 & #9 vs. #7), and pre-training the feature extractor on the same dataset yields the strongest results (#9).

We show qualitative results, comparing different feature extractors in Figure 1. Finally, we show sample results of our models after fine-tuning in Figure 4.

---

[1]DBSCAN is limited by the memory and computational complexity. We randomly select a subset of $20\,000$ pixels from the $5\,000$ input images for clustering with DBSCAN.

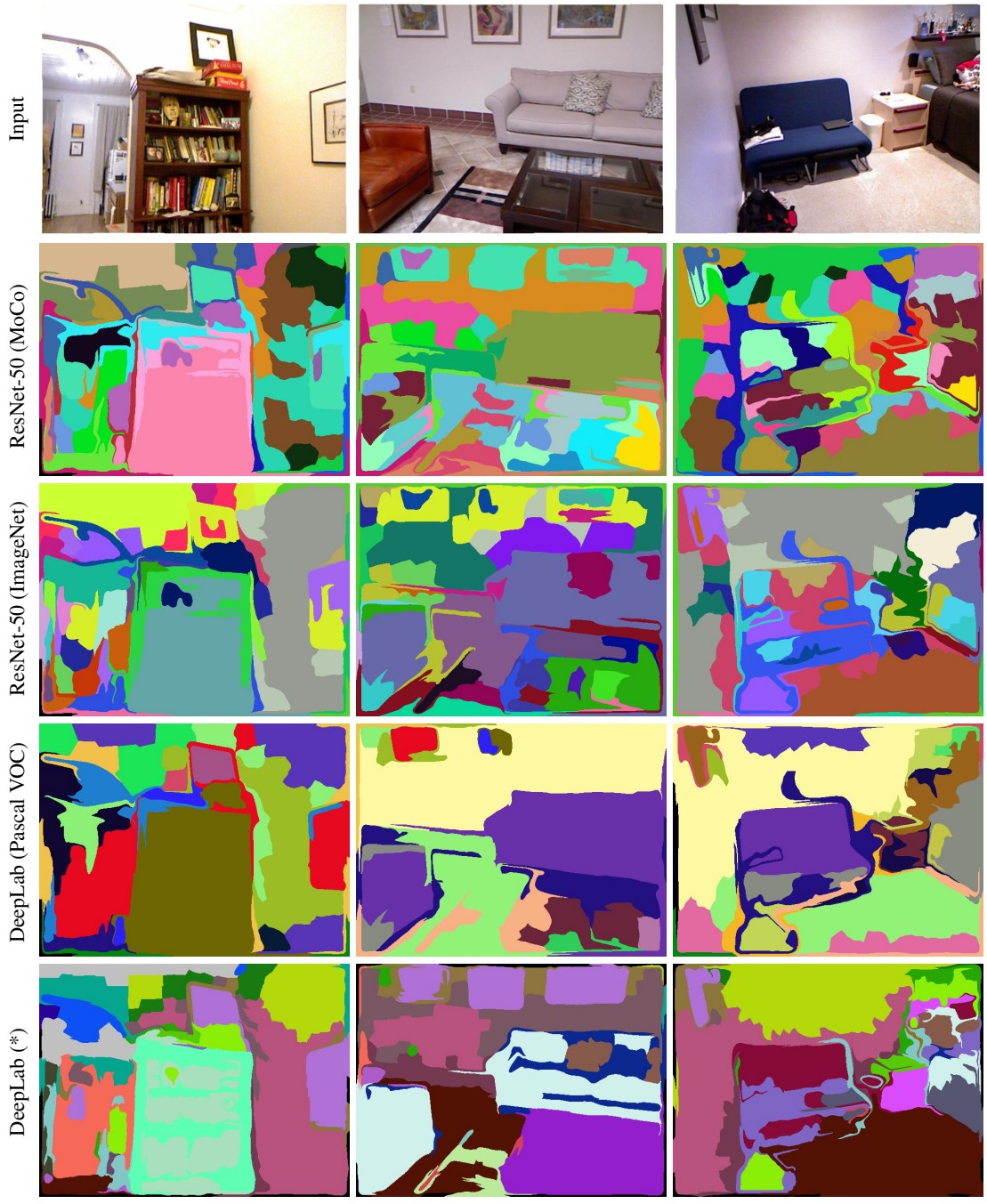

Figure 1: Qualitative comparison of various feature extractors on NYUv2. The auxiliary labels are generated using different feature extractors.

Table 2 shows the influence of using different numbers of cluster centers (and thus auxiliary labels) in K-means clustering. Corresponding qualitative results are shown in Figure 3. Tables 3-5 show ablations of various additional technical choices and hyperparameters on the Cityscapes dataset.

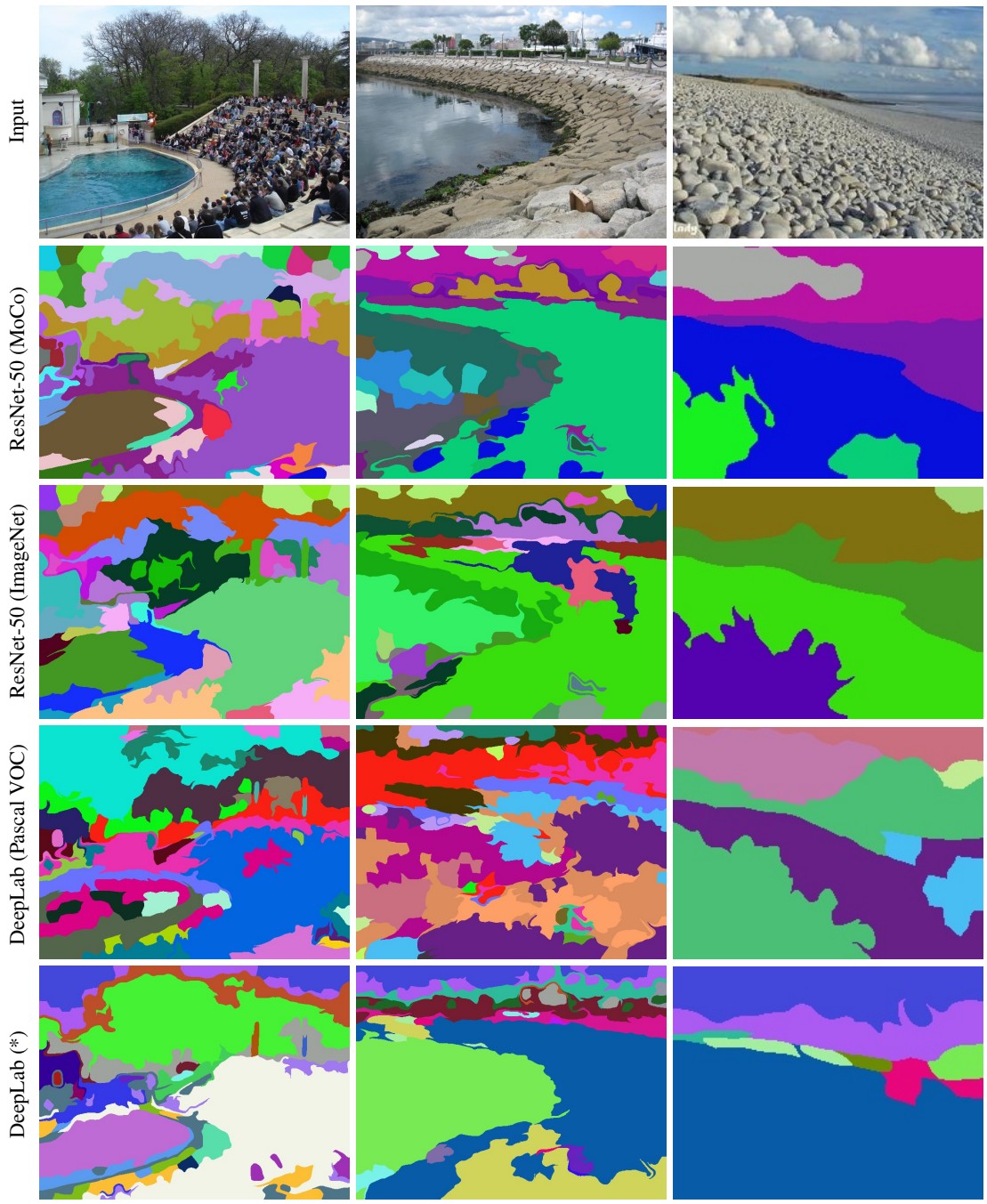

Figure 2: Qualitative comparison of various feature extractors on ADE20K. The auxiliary labels are generated using different feature extractors.

| $K$ | NYUv2 PmIoU | mIoU | Cityscapes PmIoU | mIoU |
|---|---|---|---|---|
| $1.0 \times M$ | 9.5 | 45.9 | 21.2 | 78.7 |
| $1.5 \times M$ | 9.7 | 46.0 | 21.7 | 79.2 |
| $2.0 \times M$ | 7.7 | 44.1 | 17.8 | 78.1 |

Table 2: Performance with different number of cluster centers $K$ for generating auxiliary labels. $M$ refers to the true number of classes in a dataset. Experiments were conducted with a ResNet-50 feature extractor that was trained using MoCo. Superpixel voting was enabled. We observe the best performance with $K = 1.5 \times M$.

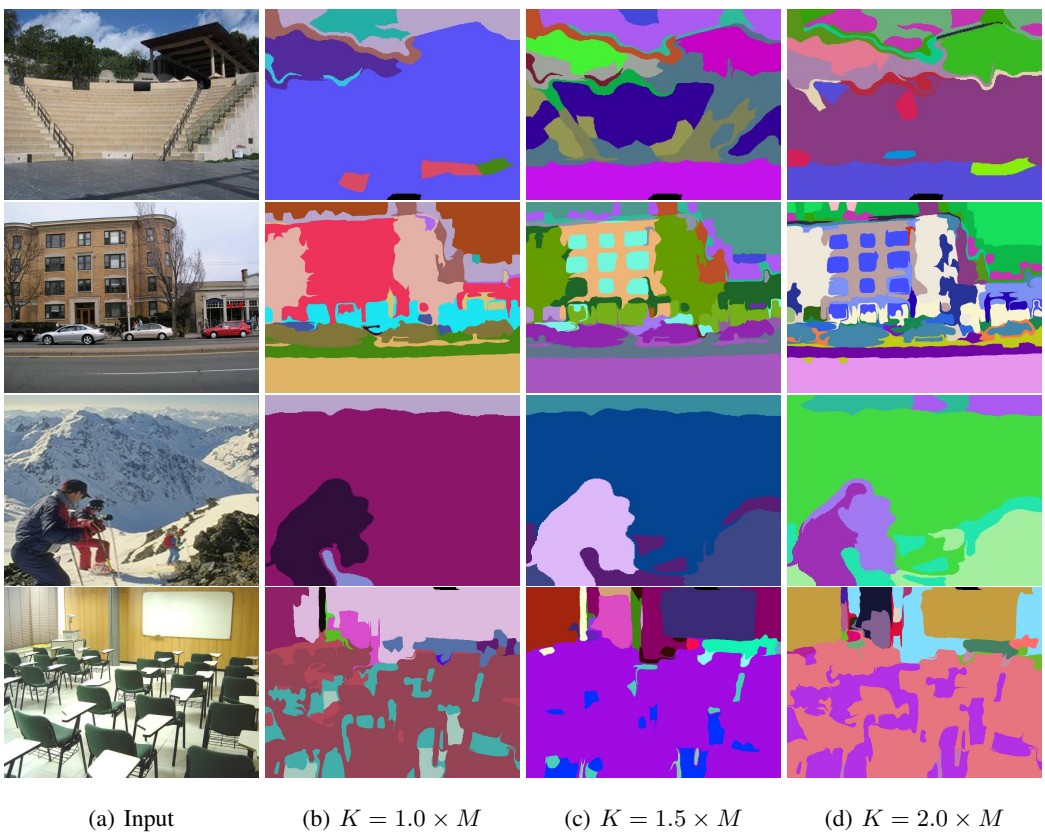

(a) Input    (b) $K = 1.0 \times M$    (c) $K = 1.5 \times M$    (d) $K = 2.0 \times M$

Figure 3: Visualization of the auxiliary labels generated with different values $K$ for clustering ($M$ is the true number of classes in the corresponding dataset).

| # superpixels | 200 | 500 | 2000 |
|---|---|---|---|
| mIoU | 81.2 | 81.2 | 81.1 |

Table 3: Effect of varying the number of superpixels per image.

| Outliers removed | 0% | 5% | 10% |
|---|---|---|---|
| mIoU | 80.9 | 81.2 | 81.0 |

Table 4: Effect of outlier removal. We observe the best performance when 5% outliers are removed.

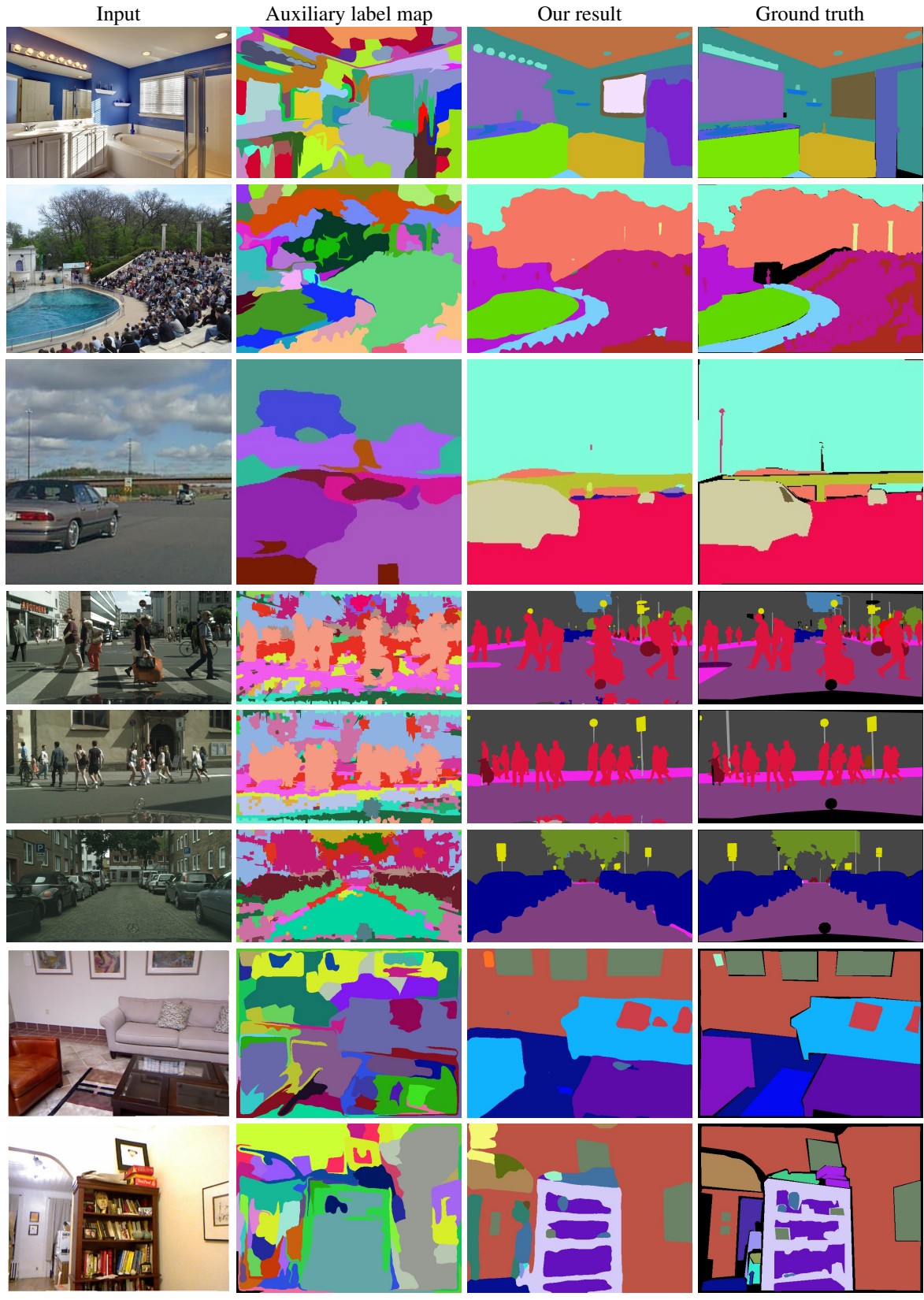

| Input | Auxiliary label map | Our result | Ground truth |

Figure 4: Comparison of auxiliary labels and results after fine-tuning.

| # images | 500 | 2000 | 5000 | 20000 | 50000 |
|----------|-----|------|------|-------|-------|
| PmIoU | 37.2 | 40.0 | 40.7 | 40.9 | 41.0 |
| mIoU | 80.7 | 81.0 | 81.2 | 81.2 | 81.3 |

Table 5: Effect of using a a varying number of images for clustering. Increasing the number of images beyond 5000 yields a minor accuracy improvement, but incurs a significantly higher computation cost.

## 1.2 Results using PSPNet

Table 6 shows a comparison to prior work for training PSPNet [4]. This complements our experiments from the main paper that show training of DeepLabV3 (Table 1 in the main paper). The results confirm that our approach is agnostic to the target architecture.

| Method | NYUv2 | | | Cityscapes | | | ADE20K | | |
|--------|-------|------|------|------------|------|------|--------|------|------|
| | 10% | 20% | 100% | 10% | 20% | 100% | 10% | 20% | 100% |
| Random initialization | - | - | 24.6 | - | - | 68.0 | - | - | 34.4 |
| ImageNet | 19.5 | 28.7 | 43.1 | 41.4 | 56.7 | 78.4 | 19.6 | 26.6 | 42.8 |
| MoCo [1] | 19.2 | 28.4 | 42.4 | 40.7 | 56.1 | 78.0 | 18.7 | 26.1 | 42.6 |
| PixPro [3] | 21.5 | 30.2 | 43.9 | 43.6 | 59.0 | 78.6 | 20.8 | 26.9 | 43.1 |
| Ours (MoCo) | 29.1 | 34.2 | 45.1 | 47.5 | 59.9 | 79.3 | 24.3 | 29.2 | 43.6 |
| Ours (DeepLabV3 (*)) | **35.4** | **39.2** | **48.8** | **51.9** | **62.3** | **81.0** | **26.0** | **30.7** | **44.1** |

Table 6: Comparison to prior work. We compare to contrastive learning methods [2, 3]. PSPNet [4] (with ResNet-50 backbone) is fine-tuned on various fractions (10%, 20%, 100%) of the target dataset. We report mean IoU after fine-tuning. We show our approach with two different feature extractors for auxiliary label generation (MoCo and DeepLabV3 fine-tuned on the target dataset, respectively).

## 1.3 Low-data regime

Table 7 shows an experiment in the low-data regime where only a small number of labeled images are available. We conduct experiments on Cityscapes and use 2000 unlabeled images together with 100 or 500 labeled images. We observe that our approach consistently outperforms ImageNet pre-training also in this low-data regime.

| # labeled images | 100 | 500 |
|------------------|-----|-----|
| Baseline (ImageNet pre-trained) | 28.1 | 55.3 |
| 2000 unlabeled images (unsupervised) | 33.1 | 57.7 |
| 2000 unlabeled images (semi-supervised) | 36.4 | 59.6 |

Table 7: Performance in the low-data regime (Cityscapes dataset). "Unsupervised" represents the result when we use MoCo [1] as the feature extractor to generate auxiliary labels. "Semi-supervised" is the result when fine-tuned DeeplabV3 (using 100/500 labeled images) is used as the feature extractor.

## 2 Additional hyperparameters

Table 8 lists the hyperparameters for the augmentations we used for generating views.

## References

[1] Xinlei Chen, Haoqi Fan, Ross Girshick, and Kaiming He. Improved baselines with momentum contrastive learning. *arXiv preprint arXiv:2003.04297*, 2020.

| Type of augmentation | Parameters |
|---|---|
| ColorJitter | (0.4, 0.4, 0.4, 0.1) |
| Grayscale | $p = 0.2$ |
| CutOut | [0, 40] |
| Scaling | [0.4, 2.5] |
| Rotation | [-60, 60] |
| Crop | [321, 361] |
| Horizontal Flip | $p = 0.5$ |

Table 8: Augmentation parameters for contrastive training. $p$ denotes the probability to convert a color image to *grayscale* or to *horizontally flip*. For *CutOut*, [0, 40] is the size in pixels of the square patch that is cut out.

[2] Kaiming He, Haoqi Fan, Yuxin Wu, Saining Xie, and Ross Girshick. Momentum contrast for unsupervised visual representation learning. In *CVPR*, 2020.

[3] Zhenda Xie, Yutong Lin, Zheng Zhang, Yue Cao, Stephen Lin, and Han Hu. Propagate yourself: Exploring pixel-level consistency for unsupervised visual representation learning. In *CVPR*, 2021.

[4] Hengshuang Zhao, Jianping Shi, Xiaojuan Qi, Xiaogang Wang, and Jiaya Jia. Pyramid scene parsing network. In *CVPR*, 2017.