# OpenReview forum: "Looking Beyond Single Images for Contrastive Semantic Segmentation Learning"
_NeurIPS.cc/2021/Conference — NeurIPS 2021 Poster_

### Official Review · Reviewer_Y4Do · 2021-07-13

**Rating:** 6
**Confidence:** 4

**Summary:**

In this paper, the authors propose a a method to learn pixel-level representations based on contrastive learning. The method rely on pre-trained feature extractor (trained unsupervised, on image-level and pixel-level supervision) to generate ‘auxiliary labels’ . The auxiliary labels, computed through K-means, are used to find positive pairs at a pixel-level. The authors show that the proposed approach achieve improved results in semantic segmentation accuracy when compared to ImageNet pretraining and other contrastive approaches.


**Ethical Concerns:**

No ethical concerns.

**Limitations And Societal Impact:**

The authors have addressed limitations and societal impact.

**Main Review:**

### Originality
The idea of learning dense (pixel-level) representation in an unsupervised way have been explored before in the literature (as mentioned by authors). Moreover, the idea of utilizing clustering in latent representation to find positive samples have also been explored (as mentioned by authors).

The difference between the proposed approach and other methods is the fact that, in this paper, the authors use a pre-trained network as feature extractor to the auxiliary labels. This poses a big issue to the model, as we can’t consider the model to be fully unsupervised if it uses a feature extractor that has been trained on labeled data.

### Quality
- The method is not fully unsupervised as the model rely on a feature extractor that has been trained with supervised data (except if considering the feature extractor trained on unsupervised data).  Therefore, it is not a fair comparison to compare the proposed approach with fully unsupervised methods (as it is done in Table 1).
- It seems that the feature extractor utilized in this method (on Table 1) was trained using supervised semantic segmentation labels. (DeepLab v3 This should be considered as the first baseline to compare the method with. The authors did not compare with this method.
- Another problem I find with this paper is the use of many different ad-hoc design choices. The performance of the method might be very dependant on many different parameters: (i) the number of clustering used, (ii) the confidence weight assigned to “type of positive pair”,  (iii) the post-processing technique, (iv) the feature extractor considered, etc.  Although some ablation study was provided, it is still very difficult to be sure where the improvement in performance really comes from.

### Clarity
The paper is in general well written and easy to follow.

### Significance
- I am not sure if any of the results shown are significant, since the proposed method rely on feature extractor that utilized labeled data (except if pretrained with MoCo).
- When training on a given dataset (eg CityScapes) and utilizing a feature extractor trained on Cityscapes as well,
- Does the feature extractor used to train the model (DeepLabV3 trained on the dataset itself) uses the whole supervised dataset? If so, it does not make sense to mention that the segmentation model was trained “only on 10%” of the data if the feature extractor was trained on the full data.

---
### Post-rebuttal updates
The authors were able to respond to most my issues and I update my final rating to 6 (specially my issue with using labeled data from another datasets in order to train the proposed model).  I believe the manuscript is slightly above acceptance rate, but would be fine it is finally rejected. My main issue is that (i) the work requires a lot of engineering steps and ad-hoc choices in order to work and, more importantly,  (ii) it utilizes labeled data in order to train the per-pixel representations.

**Time Spent Reviewing:**

3

---

> ### Author Response · Authors · 2021-08-10
> **Author Response to Reviewer Y4Do**
>
> **We provide answers to your questions below. We hope that the paper could be re-evaluated as all of the concerns were already addressed in the original submission. This includes issues on supervised vs. unsupervised learning and the DeebLab v3 baseline.**
>
>
> ### Originality
>
> 1) *The idea of learning dense (pixel-level) representation in an unsupervised way have been explored before ... we can’t consider the model to be fully unsupervised ...*
>
> Our major contribution is building cross-image contrasts in semantic segmentation. SOTAs (pixel-level) representation methods [DenseCL, PixPro] are all based on inner-image contrasts.
> We provide a flexible framework that can learn representation both in the fully unsupervised and the semi-supervised mode. We do not make any assumptions on how the feature extractor was trained. This means that it can also be trained fully unsupervised, as shown in our experiments in Table 2: ResNet50 (MoCo) hasn’t seen a single labeled sample. As such the complete pipeline has never seen a labeled sample and is thus fully unsupervised.
>
> ### Quality
>
> 1) *The method is not fully unsupervised ... Therefore, it is not a fair comparison ...*
>
> Table 2 “ResNet50 (MoCo)'' shows the fully unsupervised settings and indicates competitive performance. We additionally present results for semi-supervised learning, as to the best of our knowledge no existing approach can as flexibly leverage additional sources of data that range from coarse image-level labels to labels from a different dataset. If we missed such a baseline, we would be grateful for pointing us to it so that we can incorporate it in our evaluation.
>
> 2) *It seems that ... using supervised labels. (DeepLab v3 ... as the first baseline to compare the method with. The authors did not compare with this method.*
>
> Row 2 of Table 1, marked as “ImageNet”, is exactly the requested baseline: a DeepLabv3 architecture with a backbone that was initialized with ImageNet pre-training. The supplementary material shows a similar baseline for the PSPNet architecture. We will add a note to the paper that clarifies this.
>
>
> 3)  *Another problem ... use of many different ad-hoc design choices ... it is still very difficult to be sure where the improvement in performance really comes from.*
>
> We provide a comprehensive ablation of parameters in the paper and the supplementary material (see Tables 1 and Table 2 in the supplement for (i) and (iii), and Tables 3 and 2 in the main paper for (ii) and (iv)). Our comparisons in Table 1 are based on the best settings of the state-of-the-art methods (denseCL, PixPro) that also have the same or similar parameters. The main difference between our approach and the baselines are the cross-image correspondences which are responsible for a majority of the improvement (+3.2-5.4% mIoU).
>
> ### Significance
>
> 1) *I am not sure if any of the results shown are significant, since the proposed method rely on feature extractor that utilized labeled data (except if pretrained with MoCo).*
>
> Note that we cover a complete spectrum of methods and compare to different state of the art baselines: the fully unsupervised setting is compared to MoCov2, DenseCL and PixPro which we consistently outperform. For semi-supervised learning we compare to the state-of-the-art PseudoSeg, which we outperform, too. Our improvements are non-trivial and consistent across all experiments.
>
> 2) *When training on a given dataset (eg CityScapes) and utilizing a feature extractor trained on Cityscapes as well,*
>
> We studied the effects of different feature extractors (including unsupervised MoCo and DeepLabv3 on another smaller dataset (Pascal VOC)) in Table 2. All of them outperform the state-of-the-art significantly. Only the last row is based on the feature extractor trained on Cityscapes (the target dataset).
>
> 3) *Does the feature extractor used to train the model (DeepLabV3 trained on the dataset itself) uses the whole supervised dataset? ...*
>
> No, we did not use the whole supervised dataset in these cases. When stating “10%” of data, DeepLabv3 was also trained only on the same “10%” of labels.

---

### Official Review · Reviewer_o117 · 2021-07-16

**Rating:** 6
**Confidence:** 4

**Summary:**

Contrastive learning requires positives and negatives, and their definitions have an impact on the transfer of contrastive representations to supervised tasks. This work contributes a new definition of positives for the purpose of semantic segmentation, where the positives come not only *within* an image but *across* images. The intra-image (within) and inter-image (across) positives are derived by clustering spatially-local features across a dataset. These initial clustering features can be drawn from image-wise contrastive learning, transfer from another supervised dataset, or an oracle that is trained on some or all of the target dataset. The clustering is a straightforward but effective application of k-means; the insight is in choosing the number of desired clusters. In this work an over-clustering is desired, so that clusters correspond to "auxiliary" labels that are not necessarily the same as the supervised task labels, and more fine-grained for contrasting. This difference in labeling separates auxiliary labels from pseudo-labels, which are predictions of the supervised task labels, and underlines that auxiliary labeling can be totally unsupervised unlike pseudo-labeling. A simple metric comparing auxiliary labels to ground truth segmentations is proposed ("proxy mIoU") as a surrogate of full training as a means to cross-validate auxiliary labeling design choices.

Experiments compare the proposed method with existing contrastive learning methods (DenseContrast, MoCo, PixPro) and one recent semi-supervised semantic segmentation method (PseudoSeg, ICLR'21) on the standard datasets of Cityscapes and ADE20K plus the older, smaller dataset of NYUDv2. The downstream semantic segmentation results for all methods are measured with the DeepLabV3 architecture with a ResNet-50 backbone for standardized fair comparison. The proposed method improves on the state-of-the-art, as well as the common standard of supervised ImageNet pre-training, on all three datasets vs. both the contrastive and semi-supervised methods. The margin of improvement is largest with the least supervision, but even the fully-supervised case improves by 1-2 points. Ablations examine different choices of representation for auxiliary labeling and a variety of design choices and hyperparameters for contrastive learning.

**Ethical Concerns:**

None.

**Limitations And Societal Impact:**

The limitations and negatives are addressed. The method requires a lot of computation, as other contrastive learning methods do, and this is noted in the conclusion. This work does not have any specific negative impacts related to its particulars, but it does include a generic admission of potential bias.

**Main Review:**

This work demonstrates a role for clustering as supervision for contrastive pre-training of semantic segmentation. It mixes supervision and self-supervision to improve semantic segmentation accuracy, and unsurprisingly results improve with more supervision. That said, it does improve on other contrastive (MoCo, PixPro) and semi-supervised (pseudo-labeling) methods while surpassing fine-tuning from initialization by ImageNet classification.

Originality:
- This is the first work to propose and examine cross-image contrastive learning of spatially-local representations, to the best of my knowledge.
- There is no discussion of nor comparison with contrastive learning driven by hierarchical segmentation (Self-Supervised Visual Representation Learning from Hierarchical Grouping by Zhang & Maire, NeurIPS'20). This related work groups positives within images, and grades their relationship by distance between regions, but does not mine positives across images, as done by the submission. This comparison would gauge how much more it helps to define positives across images.
- Could the same auxiliary label approach not be of use for contrastive learning on images? It seems like this could bridge clustering-based self-supervision (see papers by Caron et al. like SwAV) and contrastive learning. Perhaps the use of clustering in this work should be related to such kinds of self-supervision, and also hybrids like PCL (Prototypical Contrastive Learning, Ji et al. ICLR'21). Coming at this from another angle, the proposed method could be cast as the local analogue of these clustering contrastive methods, where the clustered features are local instead of global.
- The proposed proxy metric is quite simple, and not so different from metrics used for box/mask proposals such as ABO (see Selective Search by Uijlings et al.).

Quality:
- Existing contrastive methods are already superior to ImageNet pre-training, so this extension of contrastive learning cannot take the credit for that. It does however improve more still, so that certainly counts. Indeed, the results show 4-6 point gaps over the second best contrastive or semi-supervised method.
- Auxiliary labels from contrastive methods are close to oracle auxiliary labels from segmentation supervision on the target dataset; see Table 2. This is a good sanity check.
- The clustering for auxiliary labeling is done on very little data (5000 images) relative to the size of these datasets. Does this matter? The experiments do not cover this choice, nor the choice of excluding outliers by cluster distance quantiles. It seems like these could be important choices for applying the proposed method.
- CityScapes and ADE20K are common and contemporary datasets for semantic segmentation. NYUDv2 is less so, as it is smaller and older with <1000 training and testing images. However, more datasets are better than fewer, so this is fine.

Clarity:
- The illustrative figures 1-4 adequately communicate the motivation, difference with prior work, and overview of the method's steps.
- Consider naming your method, for ease of reference. Perhaps "AuxCo" for auxiliary contrastive learning, as an example.
- Which feature is used for the auxiliary loss, when an intermediate feature is also used?
- The paper asserts this work is "designed from first principles" for semantic segmentation and therefore does better. This statement is unscientifically vague. Please describe the positive designs or characteristics only present in this work and/or the negative qualities hampering other work. If these cannot be specified then drop this part.

Significance:
- Annotation for segmentation is particularly burdensome, and this work improves results across all amounts of annotation over the state-of-the-art, and shows it across a sufficient set of datasets and mostly complete set of baselines.
- Contrastive baselines that make use of clustering are notably missing, such as SwAV or PCL. Without them, the results do not indicate if any kind of clustering will do, or if the proposed method is needed to make the most of contrastive pre-training for semantic segmentation.
- The experiments could be more informative about the factors underlying the method's success. For instance, basic questions like how performance varies with 1. the number of auxiliary labels, 2. the amount of unlabeled images, and 3. the source of unlabeled data (the target data, ImageNet, another segmentation dataset, ...)  are totally unanswered.
- It is unclear if the proposed method would still be of use without access to a massive amount of unlabeled data. All of the experiments assume a larger source of unlabeled data: NYUDv2 and CityScapes include 100k+ unlabeled video frames and ADE20K is supplemented with the 10ks of images in COCO. That is, how would this do when pre-training by auxiliary label contrastive learning on the standard training set alone and then fine-tuning on a subset of the labels? Although annotation is by far the more expensive effort, data collection is nevertheless not free.

Decision:
There are certainly positives, but this work is held back by the disconnect with contrastive pre-training for segmentation (see Zhang & Maire, 2020) and clustering contrastive learning (SwAV, PCL, etc.) and the lack of analysis on what is needed to make the method work. For the rebuttal, please discuss these alternative contrastive learning schemes and the significance of the proposed method in regimes with less than ~100k images. Such regimes are relevant to medical imaging and remote sensing for example.

Other Feedback
- Figure 1: Specify if the pseudo-labels and auxiliary labels are real, and derived from particular methods, or simply illustrative. (Either is fine, but it's better to be precise).
- Figure 3: There is a right arrow to nowhere from the contrastive loss. Remove it?
- Figure 5: There are different ranges for each axis, but IoU and the proposed proxy both have ranges of [0, 1], so why not plot them accordingly?
- For multi-scale feature extraction, why not choose features from different layers instead of resizing the input images? Feature pyramids are common and effective, and doing so would reduce the computation needed.

**Post Response**: The comparison with [A] is favorable and the missing analyses were adequately addressed. This settles the issues that pointed to rejection, so this paper should be considered for acceptance.

**Time Spent Reviewing:**

2.5

---

> ### Author Response · Authors · 2021-08-10
> **Author Response to Reviewer o117**
>
> ## Originality
>
> 1. *There is no discussion of nor comparison with [1]. This groups positives within images ... but does not mine positives across images ... ([1] Self-Supervised Visual Representation Learning from Hierarchical Grouping, NeurIPS'20).)*
>
> Thank you for the suggestion. We conducted this experiment and will add this comparison to the paper. Please refer to ***Table 1 in the response to Reviewer 1 (NeJV)*** for an overview of the results, which indicate a significant benefit of cross-image positive pair selection.
>
> --
>
> 2. *Could the same auxiliary label approach not be of use for contrastive learning on images? It seems like this could bridge clustering-based self-supervision ...*
>
> Thank you for this suggestion. We believe that our work indeed could be viewed as a local analogue of existing global clustering contrastive methods, but constructing a strong mathematical argument along these lines is an open question. We will explore the unification of these methods in future work.
>
> --
>
> 3.  *The proposed proxy metric is quite simple, and not so different from metrics used for box/mask proposals such as ABO (see Selective Search by Uijlings et al.).*
>
> We believe that simplicity is an important property for a metric. ABO was developed for a different task, and is unfortunately not directly applicable to semantic segmentation. We provide evidence for this in the table below:ABO is not predictive of the relative ranking of different auxiliary labeling strategies.
>
> **Table 4: Comparisons with ABO (Cityscapes)**
>
> Methods|VGG19|moco|resnet50|deeplabv3(VOC)|deeplabv3(*)|
> :-|:-:|:-:|:-:|:-:|:-:|
> ABO|28.4|26.3|27.1|25.9|45.1|
> Our PmIoU| 18.9|21.7|22.5|23.1|40.7|
> mIoU (fintuned)|78.2|79.2|79.6|80.1|81.2|
>
> ## Quality
>
> 1) *The clustering for auxiliary labeling is done on very little data (5000 images) ... the choice of excluding outliers by cluster distance quantiles ...*
>
> We found that 5000 samples strikes a good balance between computational demand of the clustering procedure and the resulting accuracy. We show an ablation of the number of samples in the Table below. We observe that by increasing the number of samples by a factor of 10, we can gain a moderate additional improvement. We will add this analysis to the supplementary material.
>
> **Table 5: Effects of using different amounts of images to train clustering algorithms.**
>
> Number of images|500|2000|5000|20000|50000|
> :-:|:-:|:-:|:-:|:-:|:-:|
> PIoU|37.2|40.0|40.7|40.9|41.0|
> mIoU (finetuned)|80.7|81.0|81.2|81.2|81.3|
>
> The choice of quantiles for outlier filtering can positively influence results. However, the overall sensitivity to this hyper-parameter remains moderate. We provide an ablation in the table below and will add this result to the supplementary material.
>
>
> **Table 6: Effects of removing outliers in positive & negative pair selection.**
>
> Remove outliers   |     0%   | 5%(default) |   10%   |
> :-:|:-:|:-:|:-:|
> mIoU (finetuned)    |    80.9    |81.2|81.0|
>
> ## Clarity
>
> 1) *Consider naming your method, for ease of reference. Perhaps "AuxCo" ...*
>
> Thanks. We will try to rename it.
>
> 2) *Which feature is used for the auxiliary loss, when an intermediate feature is also used?*
>
> We follow standard implementations: We keep the pretrained ResNet encoder up to the penultimate layer as is, attach a (trainable) fully-connected head, and pass the resulting features to the auxiliary loss.
>
> --
>
> 3) *The paper asserts this work is "designed from first principles" ... If these cannot be specified then drop this part.*
>
> We agree and will remove this claim.
>
> ## Significance
>
> 1) *Contrastive baselines that make use of clustering are notably missing, such as SwAV or PCL ...*
>
> We compare our method to PixelPro and DenseCL as they both have been shown to outperform SwAV and PCL in semantic segmentation. For completeness, we list the results for SwAV in the table below and will add them to the main paper.
>
>
> **Table 7: Comparisons with Swav and PCL**
>
> Methods|NYUv2|Cityscapes|ADE20k|
> :-|:-:|:-:|:-:|
> SWAV|41.2|77.3|43.1|
> MOCO V2|43.8|77.9|42.8|
> PCL v2|43.9|77.6|42.9|
> PixelPro|45.0|78.4|43.2|
> DenseCL|44.8|78.6|43.4|
> **Ours (unsupervised)**|**46.0**|**79.2**|**43.7**|
> **Ours (semi-supervised)**|**49.2**|**81.2**|**44.5**|
>
> --
>
> 2) *The experiments could be more informative ... like how performance varies with the number of auxiliary labels,  the amount of unlabeled images, and  the source of unlabeled data ... are totally unanswered.*
>
> These experiments are actually already in the paper and/or the supplementary material. Performance comparisons for a different number of auxiliary labels are shown in Table 2 in the supplementary material. The amount of unlabeled images is controlled for in Table 3 (see column “Data”. Observe the difference between “20k” and “128k”). The source of unlabeled data is diverse in our main experiments: NYU and Cityscapes experiments use in-domain unlabeled data, whereas the ADE20K experiments use data from an entirely different dataset. We will work on improving the text so that it becomes clear that these factors have been covered.
>
> --
>
> 3) *It is unclear if the proposed method would still be of use without access to a massive amount of unlabeled data ...  how would this do when pre-training  on the  training set then fine-tuning on a subset? ... data collection is nevertheless not free.*
>
> Thank you for this suggestion. While in many cases it is possible to collect a large set of unlabeled data cheaply, we agree that there are applications where this might not be possible. To shed some light on this issue, we applied our method to a strongly reduced training set in the table below. We used Cityscapes and sampled 2000  images from the training set (without labels) for pre-training and another 100 or 500 (with labels) for fine-tuning. We observe that our approach strongly improves on the baseline also in this setting.
>
>
> **Table 8: using fewer unlabeled images for contrastive training.**
>
> Number Finetuned/training images (labeled)|100|500|
> :-:|:-:|:-:|
> Baseline (ImageNet pretrained) |28.1|55.3|
> 2000 unlabeled images (unsupervised) |**33.1**|**57.7**|
> 2000 unlabeled images (semi-supervised)|**36.4**|**59.6**|
>  ## Other feedback
> Thank you, we will incorporate all your feedbacks.
>
> --*For multi-scale feature extraction, why not choose features from different layers instead of resizing the input images? ...*
>
> This is because 1) deep-layer features (even though in a lower resolution) contain more instance/high-level information for more accurate categorization. 2) Resizing the input images can help handle varying sizes  of the objects (especially, some far & small objects). In the experiments, concatenation of different layers performs a little worse than resizing input images. E.g. for Cityscapes, 81.0 (concatenation of layers) -> 81.2  (resizing inputs).
> "
>
> ## Decision
>
> 1) *There are certainly positives, but this work is held back by the disconnect with [1,SwAV, PCL, etc.]  ... please discuss these alternative contrastive learning schemes and the significance of the proposed method in regimes with less than ~100k images. ...*
>
> Please see our answers and additional results above. We hope that we were able to satisfactorily answer your questions and suggestions. Please let us know if you have further questions and comments.

---

> > ### Comment · Reviewer_o117 · 2021-08-21
> > **Thank you for the thorough rebuttal—I have raised my score.**
> >
> > The main negative points of my decision were lack of comparison to [A] and missing analysis of method details and suitability on different amounts of data. The comparison w.r.t. [A] is clearly favorable with 2-3 point improvement in the unsupervised case, and the observation that this work does better than MoCo v2 while [A] is only better than MoCo v1. The analyses I requested were included in the rebuttal, and would be informative if included in the supplement and referenced in the main text. In particular, I found the results in the lower data regime compelling vs. the common approach of fine-tuning from supervised pre-training. A more negative but nevertheless useful result is the flat response w.r.t. clustering over more images to make the auxiliary labels shown in Table 5 of the response. This will help guide future work.
> >
> > I have raised my score to 6 so that this submission is considered for acceptance.

---

### Official Review · Reviewer_NeJV · 2021-07-16

**Rating:** 6
**Confidence:** 3

**Summary:**

The paper presents the data collection strategy for tackling contrastive representation leaning on semantic segmentation task. The core idea is the auxiliary labels, which guide the pixel-level training sample collection across images. In addition, the authors provide a metric, i.e., PmIoU, to evaluate the quality of the generated auxiliary labels. The experimental results show that the model training guided by the auxiliary labels can be better than the previous methods.

**Limitations And Societal Impact:**

yes

**Main Review:**

[Strengths]
+ The paper is well-written and easy to follow.
+ This paper is well organized and with good descriptions.
+ The mentioned literature is sufficient.
+ This model performance is good.

[Weaknesses]
For the contributions
- Basically, the core idea is the auxiliary-label generation, which collects diverse training samples across images by leveraging K-means without annotations. However, the generation steps of feature embedding and embedding clustering are not novel compared to the previous methods, e.g., [49, 51, A].
- The metric PmIoU is designed for evaluating the quality of the auxiliary labels. However, the need for ground-truth annotations may limit its usage on datasets without annotations.

For the implementations:
- There are several implementation details described in section 3. Hence, it is not easy to realize that how many benefits are derived from the auxiliary-label generation.

For the experiments:
- It is better to discuss the relation between the superpixel size and the positive-pair partition.

Related work:
[A] Xiao Zhang, Michael Maire: Self-Supervised Visual Representation Learning from Hierarchical Grouping. NeurIPS 2020

[Justification of rating]
The proposed method is quite engineering yet has good results. My main concern is that the proposed method is lacking novelty. The proposed auxiliary-label generation for collecting diverse training samples across images has few highlights compared to previous methods.

[Post-rebuttal Updates]
The author's feedback provides more experimental results to address my concerns. Indeed, the authors demonstrate the cross-image correspondences help contrastive semantic segmentation. However, the cross-image correspondences are built upon many engineering steps and may not be easy to carry out. In sum, I agree that the work is slightly above the acceptance rate for its extension experiments, and I would like to update my final rating to “6: Marginally above the acceptance threshold.”


**Time Spent Reviewing:**

7

---

> ### Author Response · Authors · 2021-08-10
> **Author Response to Reviewer NeJV**
>
> ## Contributions
>
> 1) *Basically, the core idea is the auxiliary-label generation... However, the generation steps of feature embedding and embedding clustering are not novel compared to ... [49, 51, A].*
>
>
> Our work is similar to [49] & [51] in that all establish positive correspondences across views. These cross-image correspondences can strongly improve the performance of a contrastive learning approach since they tie together concepts that are semantically similar, but may differ in appearance substantially (see [51] and rows #1-4 of Tab.1 in our supplement). To achieve this, however, both [49] & [51] rely on dense ground truth annotations and are thus limited by the annotated portion of the training dataset and its diversity to learn the semantic similarity of apparently dissimilar concepts.
>
> The use of auxiliary labels makes our work more broadly applicable, ranging from completely unsupervised segmentation, where no labels on the target dataset are available (i.e. for pretraining), to the semi-supervised setting, when some labels - either on image-level or pixel-level - are available.
>
> While [A] also covers unsupervised semantic segmentation, it cannot benefit from any positive pairs across images as our approach (or [49,51]), since [A] creates positive correspondences only from views of the _same_ image (the case represented in Fig.2, left in our paper).
>
> To further demonstrate this crucial difference to our approach, we additionally compare ours to [A], with results shown in the table below. (As official source code is not available, we use the same dataset and network architecture as in Table 1 of [A]. ) We observe that after only a quarter of the training epochs our approach already outperforms [A] considerably. We also observe that [A] performs slightly better than the MoCo baseline while our approach dominates the even stronger MoCo2 baseline. We will add this discussion to the paper.
>
>
> **Table 1: Comparisons with reference [A]**
>
> methods|training data|val set|training epochs|mIoU |
> :-|:-:|:-:|:-:|:-:|
> [A] (unsupervised)|PASCAL+ COCO|PASCAL |80|46.5|
> Ours (unsupervised)|PASCAL+ COCO|PASCAL|**20**|**49.1**|
> Ours (semi-supervised)|PASCAL+ COCO|PASCAL|**20**|**54.3**|
>
> --
>
> 2) *The metric PmIoU is designed for evaluating the quality of the auxiliary labels. However, the need for ground-truth annotations may limit its usage on datasets without annotations.*
>
>
> The proposed PmIoU metric is predictive of relative quality even with as little as 10 annotated images. We show empirical evidence for this in the table below: We compute the PmIoU for different auxiliary labeling strategies and a varying number of fully annotated images. Even with only 10 images, PmIoU ranks the different strategies correctly with respect to their oracle performance.
> We believe that annotating 10-30 images is a reasonable effort for any practitioner, considering the time saved by alleviating the requirement to exhaustively train and fine-tune a different model for every prospective auxiliary labeling strategy.
>
>
>
> **Table 2: PIoU evaluation with only a few images (Cityscapes)**
>
> method|pixel-to-pixel|superpixel|vgg19|moco0|moco1|moco-best|Resnet-ImageNet|deeplabv3(VOC)|deeplabv3(*)|
> :-:|:-:|:-:|:-:|:-|:-:|:-:|:-:|:-:|:-:|
> PmIoU (10 images)|<0.1|<1|19.8|21.1| 21.9|23.2|24.0|24.9|44.1|
> PmIoU (30 images)| <0.1| <1| 19.3|20.2|21.5|22.4|23.2|24.0|43.6|
> PmIoU (full set: 500 images)|<0.1|<1|18.9|19.2|20.9|21.7|22.5|23.1|40.7|
> Oracle-mIoU |71.4|73.1|78.2|78.5|78.9|79.2| 79.6|80.1|81.2|
>
> ## Implementations
>
> 1)  *There are several implementation details described in section 3. Hence, it is not easy to realize that how many benefits are derived from the auxiliary-label generation.*
>
> The majority of the improvement comes from cross-image correspondences. Figure 5 of the paper and Table 1 of the supplementary material show that cross-image correspondences improve the baseline that only builds pixel-level inner-image positive pairs by about 10%. Compared to state-of-the-art methods that utilize both instance-level and pixel-level inner-image contrasts (DenseCL, PixPro), our method improves by about 4% on NYUv2 and by about 2% on Cityscapes. Note that results for the baseline and state-of-the-art are reported with the best settings for common hyper-parameters.
>
> ## Experiments
>
> 1)  *It is better to discuss the relation between the superpixel size and the positive-pair partition.*
>
> We provide an evaluation varying the size of super-pixels below. We find that the effect of different sizes is insignificant, but would happily add the analysis to the paper if reviewers agree.
>
> **Table 3: Effects of the size of the super-pixel (Cityscapes results)**
>
> Number of superpixels for each images |  200  |  500  |  2000  |
> -:|:-:|:-:|:-:|
> Finetuned mIoU after contrastive training|81.2|81.2|81.1|

---

### Author Response · Authors · 2021-08-10
**Author Response to All Reviewers**

# To All Reviewers:

We’d like to thank the reviewers for their thorough and helpful feedback. The reviewers very positively commented on multiple aspects of our paper, but also voiced various questions that predominantly concern comparisons to existing works and additional ablations of various factors. We provide detailed answers, including additional results, to the individual questions below and will incorporate this feedback in the paper.

For some common questions by reviewers, we’d like to restate that our main contribution that is responsible for the majority of the improvements is building cross-image positive contrastive pairs. Our method flexibly achieves this in different settings, ranging from fully unsupervised learning to semi-supervised learning, and consistently improves upon the closest state-of-the-art baseline in a fair setting.

However, thanks to the comments by the reviewers, we now realize that this might not be immediately obvious from the submission as the results were split up between Table 1 and Table 2.
- Table 1 compares different, representative approaches to leveraging unlabeled data for semantic segmentation, each with their best hyperparameter setting. It is a key contribution of these approaches to make best use of their unlabeled sources of data, be it annotated image classification datasets (ImageNet pretraining), vast collections of curated images (DenseContrast, MoCo), or semantic segmentation labels (PixPro, PseudoSeg).
- Table 2 shows different variants of our approach that can be compared to these different settings. We restate some important, but easy to miss, comparisons here: In the fully unsupervised setting, our result with the feature extractor “ResNet-50 (MoCo)” in Table 2 can directly be compared to the baseline “MoCo” in Table 1, where we show consistent and significant improvements (for example from 43.8% to 46.0% on NYUv2). Similarly, for the semi-supervised setting, we can directly compare “DeepLabV3 (*)” in Table 2 to “PseudoSeg” in Table 2 where we again see consistent improvements (for example from 47.1% to 49.2% on NYUv2).

We will address these clarity issues in the revision by merging the tables and adding additional discussions.

---

### Decision · Program_Chairs · 2021-09-27

**Decision:**

Accept (Poster)

**Comment:**

3 expert reviewers recommend acceptance but not without a little bit of hesitation.
The paper has a good experimental section and really drives home the point about the value of obtaining positives through correspondence across different scenes for contrastive learning (of segmentation in this case). The actual techniques proposed to achieve this were not the most elegant and there are some closely related papers, but overall seems like a valuable paper.